# Transcriptome analysis based on RNA-seq of common innate immune responses of flounder cells to IHNV, VHSV, and HIRRV

**Kwang Il Kim**[1☯], **Unn Hwa Lee**[2☯], **Miyoung Cho**[1☯], **Sung-Hee Jung**[1], **Eun Young Min**[1], **Jeong Woo Park**[2]*

**1** Pathology Research Division, National Institute of Fisheries Science, Busan, Korea, **2** Department of Biological Sciences, University of Ulsan, Ulsan, Korea

☯ These authors contributed equally to this work.
* jwpark@ulsan.ac.kr

**Data Availability Statement:** All relevant data are within the manuscript and its Supporting Information files.

## Abstract

Viral hemorrhagic septicemia virus (VHSV) and hirame rhabdovirus (HIRRV) belong to the genus *Novirhabdovirus* and are the causative agents of a serious disease in cultured flounder. However, infectious hematopoietic necrosis virus (IHNV), a prototype of the genus Novirhabdovirus, does not cause disease in flounder. To determine whether IHNV growth is restricted in flounder cells, we compared the growth of IHNV with that of VHSV and HIRRV in hirame natural embryo (HINAE) cells infected with novirhabdoviruses at 1 multiplicity of infection. Unexpectedly, we found that IHNV grew as well as VHSV and HIRRV. For successful growth in host cells, viruses modulate innate immune responses exerted by virus-infected cells. Our results suggest that IHNV, like VHSV and HIRRV, has evolved the ability to overcome the innate immune response of flounder cells. To determine the innate immune response genes of virus-infected HINAE cells which are commonly modulated by the three novirhabdoviruses, we infected HINAE cells with novirhabdoviruses at multiplicity of infection (MOI) 1 and performed an RNA sequencing-based transcriptome analysis at 24 h post-infection. We discovered ~12,500 unigenes altered by novirhabdovirus infection and found that many of these were involved in multiple cellular pathways. After novirhabdovirus infection, 170 genes involved in the innate immune response were differentially expressed compared to uninfected cells. Among them, 9 genes changed expression by more than 2-fold and were commonly modulated by all three novirhabdoviruses. Interferon regulatory factor 8 (*IRF8*), C-X-C motif chemokine receptor 1 (*CXCR1*), Toll/interleukin-1 receptor domain-containing adapter protein (*TIRAP*), cholesterol 25-hydroxylase (*CH25H*), C-X-C motif chemokine ligand 11, duplicate 5 (*CXCL11.5*), and Toll-like receptor 2 (*TLR2*) were up-regulated, whereas C-C motif chemokine receptor 6a (*CCR6a*), interleukin-12a (*IL12a*), and Toll-like receptor 1 (*TLR1*) were down-regulated. These genes have been reported to be involved in antiviral responses and, thus, their modulation may be critical for the growth of novirhabdovirus in flounder cells. This is the first report to identify innate immune response genes in flounder that are commonly modulated by IHNV, VHSV, and HIRRV. These data will provide new insights into how novirhabdoviruses survive the innate immune response of flounder cells.

**Funding:** Initials of the authors: KIK, JWP and MC Grant number: R2019055 Funder: National Institute of Fisheries Science in the Republic of Korea Initials of the authors: JWP Grant number: 2019R1A2C1006721 Funder: National Research Foundation of the Republic of Korea The funders had no role in study design, data collection, and analysis, decision to publish, or preparation of the manuscript.

**Competing interests:** The authors have declared that no competing interests exist.

## Introduction

Novirhabdovirus belongs to the family *Rhabdoviridae* and causes disease in cultured fish, resulting in significant economic losses to aquaculture industries. The genomic RNA of novirhabdoviruses has a size and structure common to the rhabdoviruses. However, unlike other rhabdoviruses, novirhabdovirus has an additional gene encoding non-structural protein (NV) located between the *G* and *L* genes and the novirhabdovirus genome encodes six viral proteins including nucleoprotein (N), phosphoprotein (P), matrix protein (M), glycoprotein (G), non-structural (NV) and RNA polymerase protein (L) in the order 3′-N-P-M-G-NV-L-5′ [1–3]. The non-structural protein NV has been reported to play a role in viral replication and in blocking the host innate immune response [4, 5].

The prototype of the genus *Novirhabdovirus* is an infectious hematopoietic necrosis virus (IHNV). In addition to IHNV, viral hemorrhagic septicemia virus (VHSV) and hirame rhabdovirus (HIRRV) are members of the genus *Novirhabdovirus*. The natural host ranges of individual novirhabdoviruses vary in breadth: while IHNV has a very narrow host range, VHSV is able to infect a wide range of species [6]. IHNV was first isolated from sockeye salmon (*Oncorhynchus nerka*) in the United States in the 1950s [7]. Even though IHNV is an acute pathogen of wild and cultured fishes in North America, Europe, and Asia [8–13], the disease caused by IHNV infection is limited to salmonid fish. On the contrary, VHSV, first isolated from rainbow trout in Europe [14], has spread to approximately 80 different fish species worldwide [8, 15, 16]. In Asia, VHSV infection was first reported in olive flounder (*Paralichthys olivaceus*) cultured in Japan in 1996 [17] and then caused disease and serious economic problems in olive flounder farming in Japan [18, 19] and Korea [20, 21]. HIRRV was first described in cultured olive flounder in Japan in the early 1980s [22], from where it gradually spread to South Korea, China [23, 24], and Europe [25] and infected a wide range of marine fishes including olive flounder, stone flounder (*Kareius bicoloratus*), black seabream (*Acanthopagrus schlegeli*), and spotted sea bass (*Lateolabrax maculatus*) [26].

Generally, the natural host range switches by viruses are reported to be rare events. In order to replicate successfully in a new host, virus must adapt to a new environment and overcome many hurdles: entering the host cell, replicating with the assistance of host factors, evading the host defense system, and spreading to other individuals [27]. However, the high mutation rates and large population sizes of RNA viruses facilitate gains in fitness and adaptation to the new environment [28]. Since the novirhabdovirus is an RNA virus and VHSV, a member of the genus novirhabdovirus, has a wide host range, it is possible that IHNV could adapt to a new host such as olive flounder, which is an important fish species in aquaculture.

The initial step of a virus adaptation to a new host is replication within the new host cells. Virus-infected cells recognize viral nucleic acids and induce the synthesis of type I interferon (IFN) (IFN-α and IFN-β) [29]. Binding of type I IFNs to the IFN receptor on the surface of neighboring cells leads to the induction of more than 300 IFN-stimulated genes (ISGs) which can restrict viral replication and, thus, play an important role in the first line of defense against infection [30–32]. The replication of novirhabdoviruses has been reported to be inhibited by the IFN system [33]. Thus, for successful replication within new host cells, virus must develop strategies to evade defense mechanisms exerted by virus-infected cells. Many viruses have developed strategies to inhibit the IFN response [31, 34] and, even though there is controversy, several studies suggest that the virulence of viruses correlates with their IFN resistance or their ability to inhibit IFN induction in infected cells [35–37].

In the present study, we investigated whether IHNV grows as efficiently as VHSV and HIRRV in hirame natural embryo (HINAE) cells. The growth of IHNV in flounder cells were found to be comparable to that of VHSV and HIRRV, indicating that IHNV, like VHSV and

HIRRV, has successfully adapted to flounder cells. We infected HINAE cells with three novirhabdoviruses at 1 multiplicity of infection (MOI) and used RNA sequencing (RNA-seq) to analyze changes in host gene expression profiles after novirhabdovirus infection, with special emphasis on the innate immune response genes that were commonly modulated by all three novirhabdoviruses. We present a list of 170 innate immune response genes in HINAE cells modulated by novirhabdovirus infection. Moreover, we identified 9 innate immune response genes commonly modulated by three novirhabdoviruses with more than 2-fold expression change. The expression of the selected genes was also validated by quantitative real-time polymerase chain reaction (qRT-PCR). These data can be used to understand innate immune responses in novirhabdovirus-infected flounder cells and provide insights into how novirhabdoviruses have adapted to flounder cells.

## Materials and methods

### Cell lines

Hirame natural embryo (HINAE) cells (RRID:CVCL_R908) from olive flounder (*Paralichthys olivaceus*) [38] were kindly provided by Dr. Ikuo Hirono (Tokyo University of Marine Science and Technology, Japan). Epithelioma papulosum cyprini (EPC) cells (ATCC Cat# CRL-2872, RRID:CVCL_4361) from fathead minnow (*Pimephales promelas*) [39] were purchased from ATCC). They were grown at 20˚C in Eagle's minimal essential medium (MEM) (GIBCO-BRL) supplemented with 10% fetal bovine serum (GIBCO-BRL).

### Viruses

IHNV (FP-RtCc 0517), HIRRV-8401H (ATCCVR-1391), and VHSV-VHS2015-5 were used in this study. The source, place of location, and year of isolation of the novirhabdoviruses used in this study are provided in Table 1. Viruses were propagated in EPC cells at 20˚C and were quantified using plaque-forming units (PFU/mL) by a standard plaque assay [40].

### Virus growth curve analysis

Infections were carried out in 25-cm$^2$ culture flasks (SPL Life Sciences) containing confluent monolayers of HINAE cells at an MOI of 0.1 and 1 PFU per cell. At the end of a 1-hour (hereafter h) absorption period at 20˚C, the inoculum was removed, and cells were washed three times with MEM containing no serum. MEM (5 mL) containing 5% fetal bovine serum (GIBCO-BRL) was then added to each culture flask, and the flasks incubated at 20˚C. Samples, consisting of 200 μL of medium, were taken at 0, 24, 72, and 120 h post-infection (p.i.) and stored at -80˚C. Viral titer was determined for all samples, in duplicate, by a plaque assay using EPC cells. Growth curves were constructed using a mean log titer for each time point.

**Table 1. Novirhabdoviruses used in this study.**

| Virus | Strain | Source | Place of isolation | Year of isolation |
|-------|--------|--------|-------------------|-------------------|
| IHNV | FP-RtCc 0517 | Rainbow trout | Korea | 2017 |
| | | (*Oncorhynchus mykiss*) | | |
| VHSV | VHS2015-5 | Olive flounder | Korea | 2015 |
| | | (*Paralichthys olivaceus*) | | |
| HIRRV | 8401 H | Olive flounder | Japan | 1984 |
| | (ATCC VR-1391) | (*Paralichthys olivaceus*) | | |

## Quantification of gene expression by qRT-PCR

Total RNA was isolated from virus-infected cells using TRIzol Reagent (Invitrogen), and 2 μg of RNA was used to generate cDNA using M-MLV reverse transcriptase (Promega). Real-time qRT-PCR was performed using SYBR Green PCR Master Mix (Qiagen) on an ABI 7500 Fast Real-Time PCR System (Applied Biosystems). The primer pairs are listed in Table 2. Following an initial 10 min denaturation/activation step at 94˚C, the mixture was subjected to 40 cycles of amplification (denaturation for 15 s at 94˚C, annealing and extension for 1 min at 60˚C). The specificity of the qRT-PCR reaction for each amplified product was verified by melting curve analysis. In each qRT-PCR run, a "no template control" was included to detect the existence of primer-dimer or contamination. β-actin gene expression is considered to be stable and we used this gene as an endogenous control gene. The relative changes in gene expression between mock and virus-infected cells were determined by $2^{-\Delta\Delta Ct}$ method using β-actin gene as the normalizing reference, where $\Delta\Delta Ct = (Ct_{target}—Ct_{reference})_{test}—(Ct_{target}—Ct_{reference})_{control}$ [41]. Each gene in each sample was normalized by subtracting the mean threshold cycle (Ct) value of the β-actin endogenous reference gene from the mean Ct value of the target gene. The fold change of each gene was obtained by calculating the difference in the normalized Ct value of target gene between mock- and virus-infected cells.

## RNA extraction and Illumina RNA-seq library preparation

HINAE cells were infected with 1 MOI of IHNV, VHSV, or HIRRV at 20˚C. Cells were collected from two independent experiments at 0 and 24 h p.i. and total RNA was isolated from virus-infected cells using TRIzol Reagent (Invitrogen). Residual DNA from each sample was removed by using the RNeasy® MinElute® Cleanup Kit (Qiagen). The total RNA in each sample were determined using a NanoDrop™ 2000 system (Thermo Fisher Scientific) and the 28S/18S ratio was estimated using an Agilent 2100 bioanalyzer (Agilent). The cDNA library was prepared with ∼1.0 μg of total RNA using a TrueSeq RNA Library Preparation Kit (Illumina) following the manufacturer's recommendations. cDNAs were amplified according to

**Table 2. PCR primers used in this study.**

| Name | Accession number | Sequences (forward and reverse, 5′ to 3′) |
|------|------------------|-------------------------------------------|
| IRF8 | XM_020098702.1 | CCAACAAGCTCTGGTGACCT, CCTGGATGTTACAGCCTTCGT |
| CXCR1 | XM_020111721.1 | GATCAGCACCAGCAAACAGA, CAGGCCATCAGCTATTGTCA |
| TIRAP | XM_020095061.1 | TGAGCTCACACCTGAGATGG, ATTGGCCTCCTCAATGTCAC |
| CH25H | XM_020111748.1 | GACTCTCAGTCTCGGCGTCT, GACAGCCACATGTTGACCAG |
| CXCL11.5 | XM_020104196.1 | TGGCTAATCCTGATGGGTTC, ACAGACTCCCGGTGTCACTC |
| TLR2 | XM_020112731.1 | TCGTCTGCACCTGTGACTTC, CTCCGTCTGTTAAACGCACA |
| CCR6A | XM_020110439 | TCACGTCTGAGGTCTTCGTG TTGTGAGAGATGAGCCGTTG |
| IL-12A | XM_020101876 | GGGGAGGATTTGCATCACTA TCTCCATGAGGGATTTGAGC |
| TLR1 | XM_020081823.1 | ACCGTCTCGCAGACTCTGTT ACGGAAACCTGGTTGTTCTG |
| β-ACTIN | XM_020109620 | GGAATCCACGAGACCACCTA, AGCACAGTGTTGGCGTACAG |

the RNA-seq protocol provided by Illumina and sequenced using an Illumina HiSeq 2500 system. Raw reads were filtered using the NGS QC Toolkit package to remove adaptor sequences and low-quality reads (base quality b < 20, read length b < 40 bp). The Trinity (r2013-11-10) pipeline program was used to assess and assemble contigs. Transcripts assembled from the total reads merged from each mRNA sample were mapped to the zebrafish genome together with the olive flounder genome using Tophat v2.0.4 with default parameters. Only those reads aligned against zebrafish or flounder genomes were considered in a differential gene expression analysis with Cuffdiff (Cufflinks v2.1.0 software) [42].

## Differentially expressed genes and enriched gene ontology and pathway analysis

Transcript abundances in reads per kilobase per million reads mapped (RPKM) were estimated using RNA-Seq by expectation maximization (RSEM) through the Trinity plug-in, run_RSEM.pl. In order to identify the differential expression patterns of transcripts the TMM-normalized fragments per kilobase of transcript per million reads mapped (FPKM = total exon fragments/[mapped reads (millions)×exon length (kb)]) matrix was used for generating heat maps under an R programming environment. Functional annotations were conducted by comparing sequences with public databases. All Illumina-assembled unigenes were compared with the NCBI non-redundant protein database (http://www.ncbi.nlm.nih.gov/) and Kyoto Encyclopedia of Genes and Genomes (KEGG) database (http://www.genome.jp/kegg) using NCBI Blast (http://www.ncbi.nlm.nih.gov/). Gene Ontology (GO) terms were assigned to each unigene based on the GO terms annotated to its corresponding homolog. Unigenes were classified according to GO terms within molecular functions, cellular components, and biological processes. Further, Unigenes were assigned to special biochemical pathways according to the KEGG database using BLASTx, followed by the retrieval of KEGG Orthology (KO) information. The Search Tool for the Retrieval of Interacting Genes (STRING) v9.1 was used to examine protein–protein interaction networks [43].

## Statistical analysis

Differences in the expression of innate immune response genes were evaluated by one-way ANOVA and two-way ANOVA using GraphPad Prism 5.0 (GraphPad Software, San Diego California USA, www.graphpad.com). A $p$ value <0.05 was considered to indicate statistical significance.

# Results

## Growth of IHNV, VHSV, and HIRRV in HINAE cells

To determine whether IHNV grows in flounder cells as efficiently as VHSV and HIRRV, we compared infectious virus yields of IHNV, VHSV, and HIRRV in a flounder cell line, HINAE, at 20˚C. HINAE cells were infected with novirhabdovirus at 0.1 and 1 MOI and culture supernatants were collected at 1, 3, and 5 days after infection. The titers of infectious virus released into the media were determined by a plaque assay. At day 1 (24 h) p.i., although the titer of IHNV was 6.3~6.9-fold lower than that of VHSV, it was 1.6~2.1-fold higher than that of HIRRV (Fig 1). At day 3 and day 5 p.i., even though the titers of the three novirhabdoviruses showed a slight difference, the viral yield of all three significantly increased and the growth curve of IHNV was similar to those of VHSV and HIRRV (Fig 1). These results suggest that IHNV is competent to replicate in flounder cells as VHSV and HIRRV.

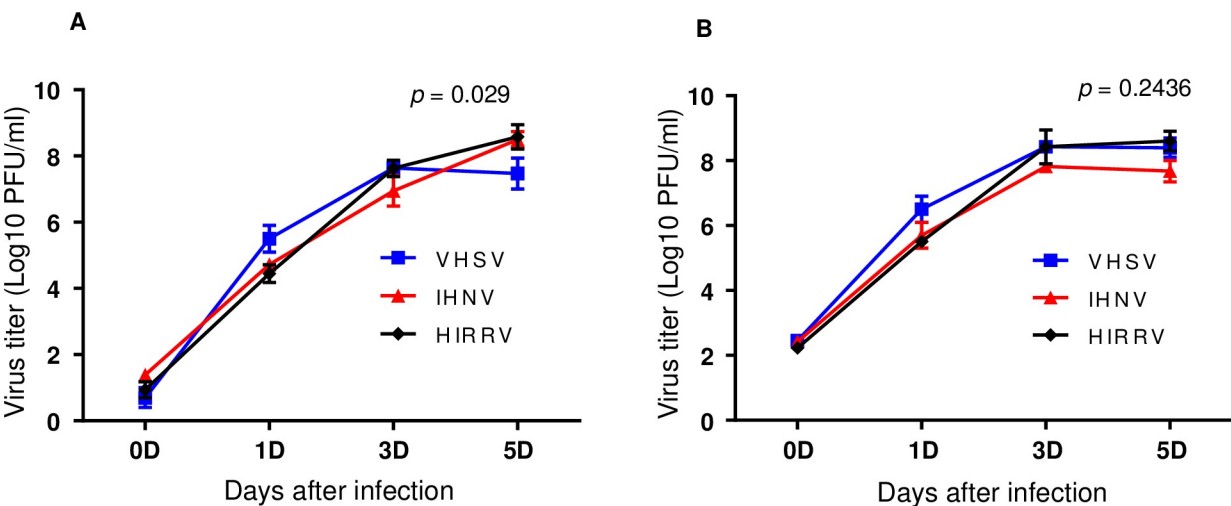

**Fig 1. Growth kinetics of IHNV, VHSV, and HIRRV in HINAE cells.** The cells were infected at a multiplicity of infection of (A) 0.1 or (B) 1 plaque-forming unit (PFU) per cell and samples of the supernatant medium were collected at the indicated time points. Viral titers were determined in duplicate by plaque assay on epithelioma papulosum cyprini (EPC) cells. Data are mean ± SD (n = 3). Two-way ANOVA. *p* values are indicated in the graphs.

## Sequencing and de novo assembly

To search for the genome-wide cell response to novirhabdovirus infection, we analyzed the transcriptomes of HINAE cells infected with IHNV, VHSV, and HIRRV by using RNA-seq and compared them with mock-treated cells. We obtained 7.1, 6.7, 8.3, and 7.2 gigabase (GB) of raw data from mock, VHSV-, IHNV-, and HIRRV-infected HINAE cells, respectively, by paired-end sequencing. After the removal of *Illumina* adaptors and filter sequences, a total of 184,875,094 cleaned reads were obtained from mock and virus-infected HINAE cells. These high-quality reads were *de novo* assembled using Trinity software v.2.2.0 [44]. Detailed information on the *de novo* transcriptome assembly is summarized in Table 3. The prediction of coding regions in the assembled transcripts was carried out using TransDecoder (implemented in Trinity software). In total, 59,980 trinity genes were found. The N50 contig length was 2,105 bp and median contig length and average contig were 427 bp and 965 bp, respectively (Table 3).

**Table 3. Summary of Illumina transcriptome sequencing and assembly for HINAE cells.**

| Summary of raw read data | Total number of raw reads | 186,432,802 |
|---|---|---|
| | Total number of clean reads | 184,875,094 |
| | Average read length after filtering (bp) | 100 |
| | Sequence quality ≥ Q30 (%) | 92.99 |
| | GC% | 50.66 |
| Summary of the assembled transcriptome | Total trinity 'genes' | 180,292 |
| | Total trinity transcripts (bp) | 302,256 |
| | GC% | 47.17 |
| | Contig N50 (bp) | 2,105 |
| | Median contig length (bp) | 427 |
| | Average contig (bp) | 965 |
| | Total assembled bases | 121,968,120 |

## Differential expression, functional annotation, and functional enrichment analysis

A total of 21,207, 20,994, and 21,335 differentially expressed genes (DEGs) were detected in IHNV-, VHSV-, and HIRRV-infected HINAE cells, respectively, when compared to mock-infected HINAE cells (S1–S3 Tables). By using gene set enrichment analysis (GSEA) [45], we identified 12,538, 12,466, and 12,595 unigenes from DEGs detected in IHNV-, VHSV-, and HIRRV-infected HINAE cells, respectively. We next conducted GO enrichment analysis to identify the major gene groups affected by infection with IHNV, VHSV, and HIRRV. GO enrichment analysis assigned these unigenes to three categories (biological process, cellular component, and molecular function) and subsequently categorized them into 53 functional groups (Fig 2 and S4 Table). Even though there was a slight difference in the number of genes in each functional group, unigenes from IHNV-, VHSV-, and HIRRV-infected HINAE cells showed a similar distribution pattern throughout the 53 functional groups (Fig 2 and S4 Table). The most prevalent GO terms were metabolic process, biological regulation, and response to stimulus in the biological process category; membrane, nucleus, and macromolecular complex in the cellular component category; and ion binding, nucleic acid binding, and protein binding in the molecular function category.

To better understand the biological functions of the unigenes, we mapped them to the referential canonical pathways in the KEGG database. The top 10 pathways based on FDR values for up- and down-regulated unigenes per each virus-infected cell are presented in Table 4. Unigenes could be mapped to 55 KEGG pathways, indicating that novirhabdovirus infection modulates the expression of cellular genes involved in a broad variety of pathways. The

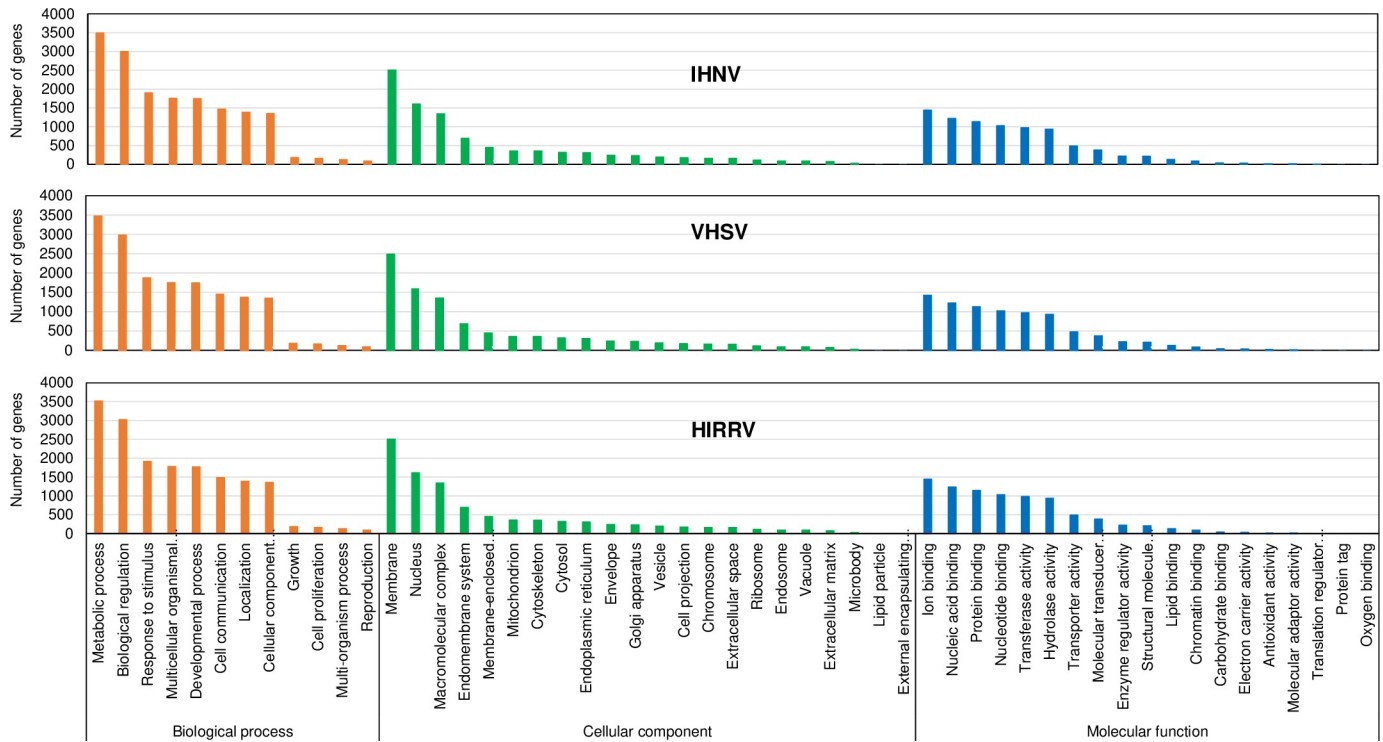

**Fig 2. Gene Ontology (GO) categories of the unigenes.** Unigenes from IHNV-, VHSV-, and HIRRV-infected HINAE cells were annotated to the 53 functional GO terms in three categories: cellular components, molecular functions, and biological process. S4 Table contains detailed information about number of genes annotated to the 53 functional GO terms.

**Table 4. KEGG pathway analysis of unigenes of IHNV-, VHSV-, and HIRRV-infected HINAE cells.**

| Virus | Up-regulated pathways | | | Down-regulated pathways | | |
|---|---|---|---|---|---|---|
| | KEGG pathway name | FDR | No. of genes | KEGG pathway name | FDR | No. of genes |
| IHNV | Notch signaling pathway | 7.78E-01 | 44 | Ribosome | 9.59E-03 | 118 |
| | Other types of O-glycan biosynthesis | 8.01E-01 | 22 | Oxidative phosphorylation | 1.09E-01 | 86 |
| | Glycosaminoglycan degradation | 8.26E-01 | 14 | Ascorbate and aldarate metabolism | 1.13E-01 | 10 |
| | Primary bile acid biosynthesis | 8.56E-01 | 9 | Folate biosynthesis | 1.24E-01 | 13 |
| | Autophagy | 8.82E-01 | 23 | Porphyrin and chlorophyll metabolism | 5.58E-01 | 21 |
| | Progesterone-mediated oocyte maturation | 8.97E-01 | 78 | Phototransduction | 8.85E-01 | 20 |
| | Glycosaminoglycan biosynthesis—keratan sulfate | 9.17E-01 | 11 | Mucin type O-glycan biosynthesis | 9.16E-01 | 18 |
| | Fatty acid elongation | 9.23E-01 | 19 | Spliceosome | 9.31E-01 | 110 |
| | Phenylalanine metabolism | 9.25E-01 | 11 | ABC transporters | 9.35E-01 | 32 |
| | N-Glycan biosynthesis | 9.33E-01 | 41 | Pantothenate and CoA biosynthesis | 9.64E-01 | 11 |
| VHSV | Neuroactive ligand-receptor interaction | 8.63E-03 | 179 | Ribosome biogenesis in eukaryotes | 0E+00 | 61 |
| | Cell adhesion molecules | 1.04E-01 | 75 | Spliceosome | 0E+00 | 110 |
| | Drug metabolism—other enzymes | 1.98E-01 | 20 | DNA replication | 9.95E-04 | 32 |
| | Calcium signaling pathway | 2.02E-01 | 158 | Glycosylphosphatidylinositol-anchor biosynthesis | 1.24E-03 | 20 |
| | Cytokine-cytokine receptor interaction | 2.13E-01 | 96 | Nucleotide excision repair | 1.52E-03 | 42 |
| | Adrenergic signaling in cardiomyocytes | 2.20E-01 | 128 | Base excision repair | 1.66E-03 | 29 |
| | Tryptophan metabolism | 2.33E-01 | 32 | Basal transcription factors | 2.52E-03 | 37 |
| | ABC transporters | 2.42E-01 | 34 | Fanconi anemia pathway | 2.80E-03 | 44 |
| | Ether lipid metabolism | 2.43E-01 | 26 | Peroxisome | 6.13E-03 | 67 |
| | Tyrosine metabolism | 2.49E-01 | 22 | RNA transport | 7.97E-03 | 132 |
| HIRRV | Taurine and hypotaurine metabolism | 9.78E-01 | 6 | Ribosome | 3.47E-01 | 118 |
| | RNA polymerase | 9.89E-01 | 29 | Folate biosynthesis | 5.37E-01 | 13 |
| | Protein export | 9.97E-01 | 19 | Arachidonic acid metabolism | 6.23E-01 | 28 |
| | Purine metabolism | 9.97E-01 | 140 | Nicotinate and nicotinamide metabolism | 7.38E-01 | 22 |
| | Insulin signaling pathway | 9.98E-01 | 124 | Mismatch repair | 9.69E-01 | 21 |
| | Herpes simplex infection | 9.99E-01 | 124 | Protein processing in endoplasmic reticulum | 9.97E-01 | 134 |
| | Progesterone-mediated oocyte maturation | 1.00E+00 | 79 | Oxidative phosphorylation | 1.00E+00 | 86 |
| | Oocyte meiosis | 1.00E+00 | 96 | Tight junction | 1.00E+00 | 113 |
| | Adherens junction | 1.00E+00 | 65 | Neuroactive ligand-receptor interaction | 1.00E+00 | 181 |
| | FoxO signaling pathway | 1.00E+00 | 131 | Pantothenate and CoA biosynthesis | 1.00E+00 | 11 |

distribution of gene pathways among unigenes from IHNV-, VHSV-, and HIRRV-infected cells was quite different. However, genes related to signaling pathways were commonly up-regulated but those related to ribosome and repair were commonly down-regulated. This analysis showed enriched pathways in flounder cells associated with novirhabdovirus infection.

## Modulation of the innate immune response by the three novirhabdoviruses

The growth of a virus in host cells depends on its ability to modulate the innate immune response in infected cells [35–37]. We found that IHNV grew as well as VHSV and HIRRV in flounder cells, suggesting that IHNV can modulate the key innate immune response of flounder cells as efficiently as VHSV and HIRRV. To determine the innate immune responses of flounder cells which are commonly modulated by three novirhabdoviruses, we analyzed the expression changes in innate immune response genes of HINAE cells infected with IHNV, VHSV, and HIRRV for 24 h and compared them with mock-infected control cells.

From the RNA-seq analysis, we identified 170 DEGs involved in innate immune response at 24 h p.i. (Fig 3A and S5 Table). Of the 170 DEGs, 22, 66, and 28 genes showed more than a

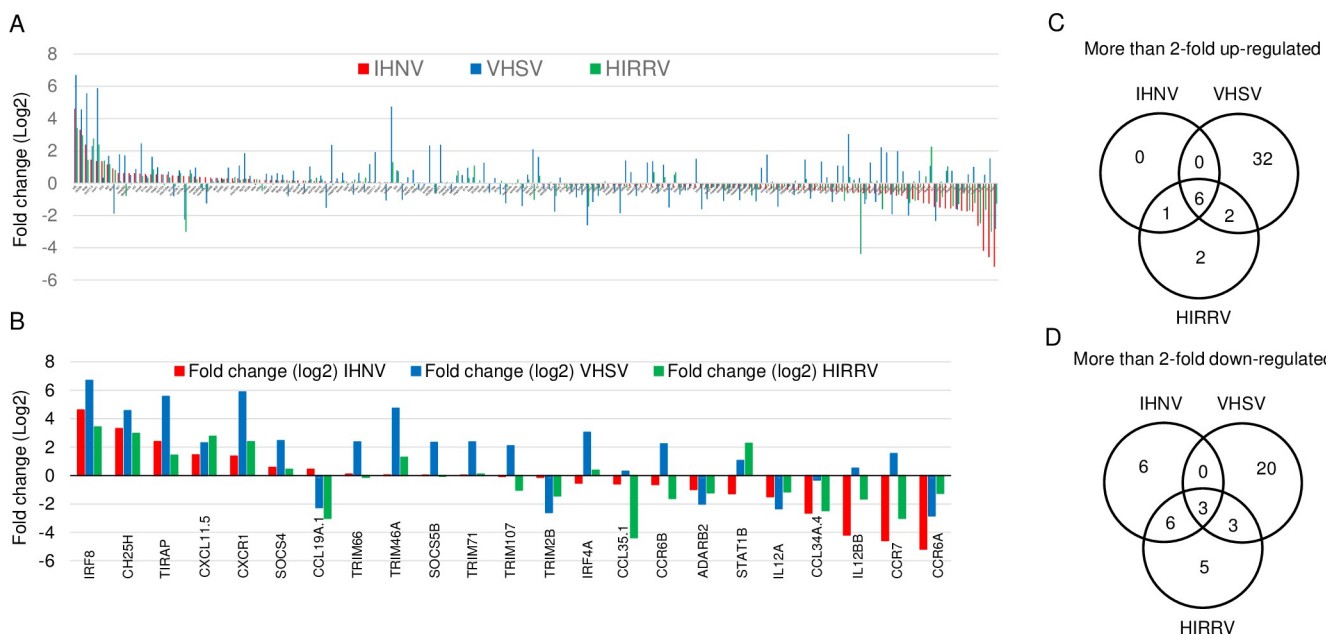

**Fig 3. Effect of novirhabdovirus infection on host gene expression of HINAE cells.** (A) Fold changes of (A) 170 innate immune response genes and (B) 23 innate immune response genes with more than 4-fold change in HINAE cells at 24 h after 1 multiplicity of infection (MOI) with IHNV, VHSV, or HIRRV. S5 Table contains detailed information about gene name and fold change value for each gene. (C & D) Numbers in the Venn diagrams (http://bioinfogp.cnb. csic.es/tools/venny/index.html) represent the number of cellular genes with more than two-fold (C) up-regulated and (D) down-regulated in IHNV-, VHSV- , and HIRRV-infected cells at 24 h post-infection.

2-fold expression change in HINAE cells after infection with IHNV, VHSV, and HIRRV, respectively (Fig 3A and S5 Table). In particular, 9 genes were commonly modulated with more than a 2-fold change by the three novirhabdoviruses: 6 genes, interferon regulatory factor 8 (*IRF8*), C-X-C motif chemokine receptor 1 (*CXCR1*), TIR domain containing adaptor protein (*TIRAP*), cholesterol 25-hydroxylase (*CH25H*), chemokine (C-X-C motif) ligand 11, duplicate 5 (*CXCL11.5*), and Toll-like receptor 2 (*TLR2*), were increased; 3 genes, chemokine (C-C motif) receptor 6a (*CCR6a*), interleukin 12a (*IL12a*), and Toll-like receptor 1 (*TLR1*), were decreased (Fig 3B and 3C, and Table 5). These results suggest that modulation of these genes may be essential for the replication of the three novirhabdoviruses in flounder cells. On the other hand, 65 genes were species-specific: 6 IHNV-specific genes, 52 VHSV-specific genes, and 7 HIRRV-specific genes (Fig 3B and 3C, and Table 5). These species-specific genes may be involved in the species-specific pathogenicity of the novirhabdoviruses.

To confirm these results, we selected 9 commonly modulated genes and determined by qRT-PCR their expression levels in novirhabdovirus-infected HINAE cells. Consistent with RNA-seq results, we found a significant increase in gene expression for *IRF8*, *CXCR1*, *TIRAP*, *CH25H*, *CXCL11.5*, and *TLR2* and a significant decrease in gene expression for *CCR6a*, *IL-12A*, and *TLR1* at 24 h after infection with the three novirhabdoviruses (Fig 4 and S6 Table). These results suggest that the induction of these genes is a common event in HINAE cells infected by IHNV, VHSV, and HIRRV.

## Protein–protein interaction network analysis

The biological associations among the innate immune response genes modulated by IHNV, VHSV, and HIRRV with more than 2-fold change were investigated using STRING software. The predicted protein–protein associations were queried through a vast number of databases

**Table 5. Genes modulated more than 2-fold in HINAE cells at 24 h after infection with IHNV, VHSV, or HIRRV.**

| | Virus specificity | Gene name |
|---|---|---|
| >2-fold increase | Common to IHNV, VHSV, and HIRRV | IRF8, CXCR1, TIRAP, CH25H, CXCL11.5, TLR2 |
| | IHNV | -* |
| | VHSV | IRF4A, SOCS4, TRIM71, TRIM66, SOCS5B, CCR6B, TRIM107, TRIM46B, CXCL11.7, MAPK12B, ISG15, TLR7, CXCR4A, CASP8L2, TRIM25, TRIM35-12, CCR7, IRF3, MOV10A, CRFB2, IL1B, IRF1A, TRIM8B, TRIM62, TRIM36, TRIM8A, TRIM110, TRIM24, MHC1UBA, CCL20A.3, ADARB1B, CCL36.1 |
| | HIRRV | FOSAB, CXCL19 |
| >2-fold decrease | Common to IHNV, VHSV, and HIRRV | CCR6A, IL12A, TLR1 |
| | IHNV | CXCR5, CXCL19, CCR10, STAT1B, IL12RB2, IL10 |
| | VHSV | CXCL14, IFIT16, TRIM55A, IL17C, TRIM45, ISG20L2, IFNGR1, RNASEH1, TRIM32, SOCS7, MOV10B.2, TLR3, TRIM2A, TRIM105, SOCS1A, SOCS2, TRIM33L, SOCS9, RNASEH2B, CDC40 |
| | HIRRV | CCL35.1, CCR6B, CCL20A.3, TRIM109, TRIM107 |

IRF8, interferon response factor 8; CXCR1, C-X-C motif chemokine receptor 1; TIRAP, TIR domain containing adaptor protein; CH25H, cholesterol 25-hydroxylase; CXCL11.5, chemokine (C-X-C motif) ligand 11, duplicate 5; TLR2, Toll-like receptor 2; IRF4A, interferon response factor 4a; SOCS4, Suppressor Of Cytokine Signaling 4; TRIM71, Tripartite Motif Containing 71; TRIM66, Tripartite Motif Containing 71; SOCS5B, Suppressor Of Cytokine Signaling 5b; CCR6B, chemokine (C-C motif) receptor 6b; TRIM107, Tripartite Motif Containing 107; TRIM46B, Tripartite Motif Containing 46b; CXCL11.7, chemokine (C-X-C motif) ligand 11, duplicate 7; MAPK12B, mitogen-activated protein kinase 12b; ISG15, interferon-stimulated gene 15; TLR7, Toll-like receptor 7; CXCR4A, chemokine (C-X-C motif) receptor 4a; CASP8L2, caspase 8, apoptosis-related cysteine peptidase, like 2; TRIM25, Tripartite Motif Containing 25; TRIM35-12, Tripartite Motif Containing 71; CCR7, chemokine (C-C motif) receptor 7; IRF3, interferon response factor 3; MOV10A, Moloney leukemia virus 10a; CRFB2, cytokine receptor family member B2; IL1B, interleukin 1b; IRF1A, interferon response factor 1a; TRIM8B, Tripartite Motif Containing 8b; TRIM62, Tripartite Motif Containing 62; TRIM36, Tripartite Motif Containing 36; TRIM8A, Tripartite Motif Containing 8a; TRIM110, Tripartite Motif Containing 110; TRIM24, Tripartite Motif Containing 24; MHC1UBA, Major histocompatibility complex class I UBA; CCL20A.3, chemokine (C-C motif) ligand 20a, duplicate 3; ADARB1B, adenosine deaminase RNA specific B1b; CCL36.1, chemokine (C-C motif) ligand 36, duplicate 1; FOSAB, v-fos FBJ murine osteosarcoma viral oncogene homolog Ab; CXCL19, chemokine (C-X-C motif) ligand 19; CCR6A, chemokine (C-C motif) receptor 6a; IL12A, interleukin 12a; TLR1, Toll-like receptor 1; CXCR5, chemokine (C-X-C motif) receptor 4a; CCR10, chemokine (C-C motif) receptor 10; STAT1B, Signal transducer and activator of transcription 1b; IL12RB2, interleukin 12 receptor subunit beta 2; IL10, interleukin 10; CXCL14, chemokine (C-X-C motif) ligand 14; IFIT16, interferon-induced protein with tetratricopeptide repeats 16; TRIM55A, Tripartite Motif Containing 55a; IL17C, interleukin 17c; TRIM45, Tripartite Motif Containing 45; ISG20L2, interferon stimulated exonuclease gene 20 like 2; IFNGR1, Interferon gamma receptor 1; RNASEH1, Ribonuclease H1; TRIM32, Tripartite Motif Containing 32; SOCS7, Suppressor Of Cytokine Signaling 7; MOV10B.2, Moloney leukemia virus 10b, tandem duplicate 2; TLR3, Toll-like receptor 3; TRIM2A, Tripartite Motif Containing 2a; TRIM105, Tripartite Motif Containing 105; SOCS1A, Suppressor Of Cytokine Signaling 1a; SOCS2, Suppressor Of Cytokine Signaling 2; TRIM33L, Tripartite motif-containing 33,-like; SOCS9, Suppressor Of Cytokine Signaling 9; RNASEH2B, Ribonuclease H2, subunit B; CDC40, Cell Division Cycle 40; CCL35.1, chemokine (C-C motif) ligand 35, duplicate 1; TRIM109, Tripartite Motif Containing 109.

*, no genes specific to IHNV.

derived in different ways (e.g. experimentally determined interactions, protein neighborhood data, or data acquired via text mining). Interestingly, most of the innate immune response genes modulated by the three novirhabdoviruses were found to be connected with *TLR1/TLR2/TIRAP* genes and *CXCR1/CCR6a* genes (Fig 5).

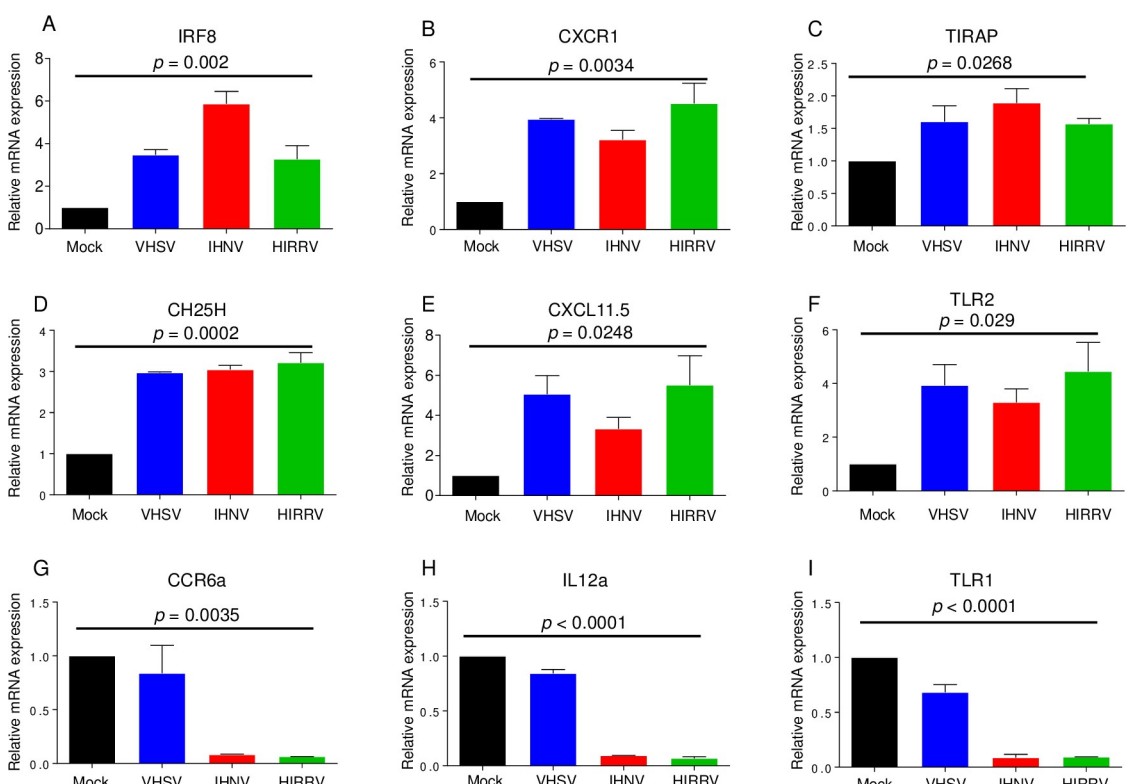

**Fig 4. Confirmation of RNA-seq data by RT-qPCR after novirhabdovirus infection.** Gene expression for the indicated genes was assessed by qRT-PCR from HINAE cells after 1 multiplicity of infection (MOI) with IHNV, VHSV, or HIRRV at 20°C for 12 h and 24 h. The *β*-actin gene was used as a reference to normalize the data. Expression values are shown as fold change compared to untreated cells. Data are mean ± SEM (*n* = 6). One-way ANOVA. *p* values are indicated in the graphs. S6 Table contains detailed information about Ct values of each gene.

## Discussion

In this study, we first investigated whether IHNV is competent to replicate in flounder cells as VHSV and HIRRV. We compared the production of infectious viral progeny by the three novirhabdoviruses in flounder cells and found that the production of IHNV was similar to that of VHSV and HIRRV. These results suggest that IHNV is competent to replicate in flounder cells.

Novirhabdovirus infection induces a rapid IFN and ISG response [46–49] and the pre-activated IFN and ISG response inhibits the replication of novirhabdoviruses [50–53]. For successful replication within host cells, novirhabdovirus must possess a mechanism to inhibit the IFN response of host cells. To better understand the underlying mechanisms of the IHNV-mediated modulation of IFN responses in flounder cells, we compared the whole genome transcriptome analysis of flounder cells infected with three novirhabdoviruses at 24 h p.i. using RNA-seq. We first determined the cellular transcriptome profile to investigate the changes in host expression during infection. Infection by the novirhabdoviruses induced massive changes (about 12,500 unigenes compared to the transcriptome of uninfected control cells) in the flounder cell transcriptome. GO enrichment analysis revealed that about 12,500 unigenes from novirhabdovirus-infected flounder cells could be assigned to 53 functional groups and that unigenes from IHNV-, VHSV-, and HIRRV-infected cells similarly distributed to these 53 functional groups. GO enrichment analysis also suggest that HINAE transcriptome is similarly

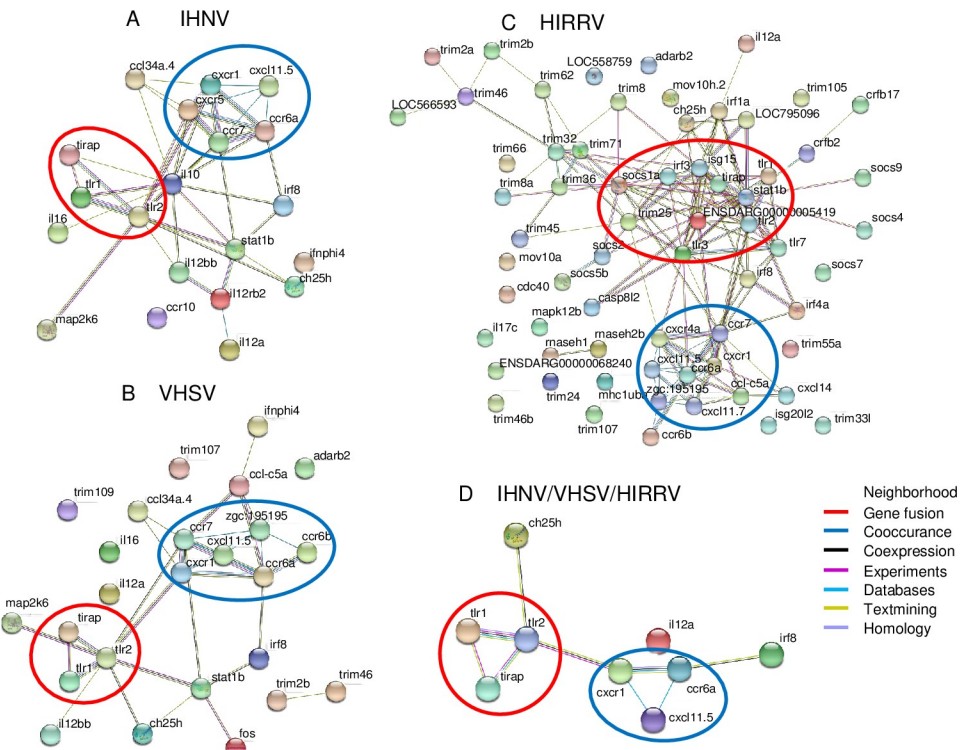

**Fig 5. Interaction networks of innate immune response genes modulated by novirhabdovirus infection by STRING analysis.** Interaction networks of the associated proteins were found among 22, 66, and 28 genes with more than 2-fold expression change in HINAE cells after infection with (A) IHNV, (B) VHSV, and (C) HIRRV, respectively. (D) Protein interaction network analysis of 9 genes commonly modulated by IHNV, VHSV, and HIRRV. The interaction networks included two highly connected protein nodes: red circle, node consisted of TLR1/TLR2/TIRAP proteins; blue circle, node consisted of CXCR1/CCR6a proteins. Different line colors represent the types of evidence for the association.

modulated by the three novirhabdoviruses and that a large part of the modulated HINAE transcriptome is devoted to the control of the main metabolic processes and to binding (ion, nucleic acid, and protein). Contrary to the result of GO enrichment analysis, unigenes from the three novirhabdoviruses quite differently distributed to 55 KEGG pathways. Even though the distribution pattern of the unigenes affected by the three novirhabdoviruses was different we could detect a common trend: genes related to signaling pathway were up-regulated and those related to ribosome and repair were down-regulated.

In HINAE cells, we found that novirhabdovirus infection altered the expression of 170 genes related to the IFN response at 24 h p.i. Among 170 genes, 6 genes (*IRF8*, *CXCR1*, *TIRAP*, *CH25H*, *CXCL11.5*, and *TLR2*) were up-regulated more than two-fold by three novirhabdoviruses. In particular, there was a 103-fold, 24-fold, and 10-fold increase of *IRF8* expression in VHSV-, IHNV-, and HIRRV-infected cells, respectively. *IRF8*, a transcription factor, is able to either inhibit [54, 55] or induce IFN gene transcription [56–58] depending on cell type and conditions. The role of IRF8 in antiviral immunity also depends on cell type and condition. For example, IRF8 in cluster of differentiation 8[+] (CD8[+]) T cells suppresses the activation and expansion of CD8[+] T cells in herpes simplex virus 1 (HSV-1) infected mice, which reduces antiviral immunity [59]. On the other hand, IRF8 in human monocytes is involved in the rapid induction of IFN-β after exposure to Sendai virus [57]. In addition, IRF8 in mouse natural killer cells exposed to cytomegalovirus facilitates the proliferation of virus-specific natural

killer (NK) cells and enhances NK cell mediated antiviral immunity [60]. Even though it has been reported that *IRF8* is induced by viruses in olive flounder [61], the role of *IRF8* in antiviral immunity in fish species has not yet been determined.

As in *IRF8*, the other genes in the group of six that were up-regulated, *CXCR1*, *TIRAP*, *CH25H*, *CXCL11.5*, and *TLR2*, have been reported to be induced by virus infection and have both antiviral and proviral functions. CXCR1, a member of the G-protein-coupled receptor family is a chemotactic factor [62]. CXCR1 and its ligand C-X-C motif chemokine ligand 8 (CXCL8)/interleukin 8 (IL-8) have been reported in multiple fish species [63] and fish CXCL8/IL-8 is also induced in spleen upon infection with VHSV [64] and IHNV [46]. Co-injection of IL-8 with VHSV G-encoding DNA vaccine in rainbow trout enhanced the production of proinflammatory cytokine [65]. This suggests the possibility that CXCR1 may have antiviral function by enhancing recruitment of immune cells to the virus infection site. However, there are several reports indicating that CXCR1 has proviral function. CXCR1 ligand IL-8 induced by virus infection enhances the replication of several viruses including encephalomyocarditis virus (EMCV) [66] and cytomegalovirus (CMV) [67].

TIRAP is an adaptor protein in the TLR signaling pathway that functions downstream of TLRs [68, 69]. Recruitment of TIRAP to TLRs induces antiviral innate immune response through the activation of nuclear factor (NF)-κB and IRFs. For example, upon stimulation by herpes simplex virus (HSV) and influenza viruses, TIRAP in association with TLR9 and TLR7 induces expression of IFN-α which plays an antiviral role [70]. In macrophages after exposure to HSV-1, TIRAP is involved in the induction of IL-15 [71], which is essential for the generation, activation, and proliferation of NK and natural killer T (NKT) cells [72]. On the contrary, it has been reported that NF-κB downstream of TIRAP can enhance HSV-1 gene expression and increase viral yields [73].

*CH25H* is rapidly induced by various TLR ligands and IFN molecules [74]. CH25H catalyzes the formation of 25-hydroxycholesterol (25HC) which exerts an antiviral activity through both 25HC-dependent and independent events. The 25HC protein impairs viral entry at the virus-cell membrane fusion step by inducing cellular membrane changes [75] and, thus, suppresses infection of enveloped viruses including vesicular stomatitis virus (VSV) and human immunodeficiency virus type 1 (HIV-1) [75, 76]. Alternatively, CH25H has been reported to interact with the nonstructural protein 5A (NS5A) protein of hepatitis C virus (HCV) and inhibit HCV replication [77]. In zebrafish, CH25H is induced by Spring Viremia Carp Virus (SVCV), inhibits VHSV replication, and protects zebrafish larvae from SVCV infection [78]. In addition, CH25H is induced by Singapore grouper iridovirus (SGIV) and red-spotted grouper nervous necrosis virus (RGNNV) in grouper cells and inhibits the entry and replication of these viruses [79].

TLR2 is a member of the TLR family. TLR2 stimulated by ligand activates NF-κB to induce production of cytokines, including IFN-β and inflammatory cytokines, and chemokines [80]. The major TLR2 ligands are lipoproteins, highly expressed in the outer membrane of Gram-positive bacteria [81]. TLR2 can recognize viral components such as envelope glycoproteins [82]. Evidence suggests that TLR2 plays antiviral roles: TLR2 stimulated by vaccinia virus and murine CMV induces IFN-β production in monocytes [83]; TLR2 knockout impairs NK cell activity and increases the CMV load in mice [84]. In fish cells, TLR2 expression shows a negative correlation with the transcription level of VHSV in the kidney of olive flounder [85]. In turbot, VHSV infection significantly down-regulates vaccine-induced TLR2 expression [86]. By contrast with these findings, in this study, we found that the infection of novirhabdoviruses increased the expression of *TLR2* in flounder cells.

In this study, we found that *IRF8*, *CXCR1*, *TIRAP*, *CH25H*, *CXCL11.5*, and *TLR2* are commonly induced by three novirhabdoviruses in flounder cells, indicating that the upregulation

of these six genes is essential in novirhabdovirus-infected flounder cells. It is highly possible that these six genes may have an antiviral function in other virus-infected cells and play a role in the suppression of viral growth. However, it is also possible that these genes could mediate signaling pathways that may be required for the efficient growth of novirhabdoviruses in flounder cells. More work should be done to determine whether these six genes induced by the three novirhabdoviruses in flounder cells play an antiviral role or support the growth of novirhabdovirus in flounder cells.

On the contrary, the expression of three innate immune response genes, including *CCR6a*, *IL12a*, and *TLR1* was down-regulated more than two-fold by three novirhabdoviruses. CCR6a is a CC chemokine receptor found in teleost fish [87, 88]. It has been reported that CCR6a is induced in olive flounder after infection with *Vibrio anguillarium* [88], indicating that CCR6a plays a role in the immune response against infection. However, its role in infection has not yet been determined. CCR6a in teleost fish shows the highest identity with human and mouse CCR6 [87]. C-C motif chemokine ligand 20 (CCL20) is the only confirmed chemokine ligand of CCR6 [89]. CCR6 is induced by virus infection and modulates the migration of immune cells. For example, Kaposi's sarcoma-associated herpesvirus (KSHV) induces CCR6 expression in human umbilical vein endothelial cells, which may contribute to the recruitment of dendritic cells (DCs) and lymphocytes into the Kaposi's sarcoma lesion [90]. Migration of immune cells induced by CCR6 and CCL20 signaling may be beneficial to viral growth since neutralization of CCL20 or CCR6 depletion decreases the migration of conventional DCs to the lung and enhances clearance of respiratory syncytial virus (RSV) [91].

IL12a is a subunit of the cytokine IL-12, a heterodimer composed of IL-12a and IL-12b. IL-12 is a proinflammatory cytokine that promotes the differentiation of type 1 T helper (Th1) cells. Accumulating evidence suggests that IL-12 plays a role in the efficient immune response to virus infections. It enhances the production of antiviral cytokines and amplifies the cytotoxicity of cytotoxic T lymphocytes [92] which can kill hepatitis B virus (HBV)-infected hepatocytes [93]. IL-12 induces the production of TNF-α and INFα/ß which inhibits the growth of hepatitis B virus in liver and kidney cells [94]. To escape from the immune response, HBV down-regulates IL-12a expression after infection into host cells [95].

TLR1 in association with TLR2 can recognize envelope glycoprotein gB and gH of human cytomegalovirus (HCMV) [96]. In addition, *TLR1* expression is induced by influenza A virus and Sendai virus in human macrophages [97]. This suggests that TLR1 has an antiviral role. TLR1 has been identified in olive flounder [98]. In addition, TLR1 expression is induced by VHSV infection in fish cells and may play an antiviral role [99]. Here we found that the expression of *CCR6a*, *IL12a*, and *TLR1* was down-regulated by the three novirhabdoviruses in flounder cells. When considering the antiviral function of these genes, their inhibition seems to be essential for the efficient growth of novirhabdoviruses in flounder cells. However, the role of these genes in innate immunity against novirhabdovirus infection in fish cells remains to be determined.

Beside the above-mentioned common genes, our results indicated that three novirhabdoviruses affected different sets of innate immune response genes in cells. Of the 170 innate immune response genes, VHSV infection affected the expression of 66 genes more than two-fold, while IHNV and HIRRV altered the expression of 22 and 28 genes more than two-fold, respectively. These indicate that flounder cells respond more actively to VHSV infection than to IHNV or HIRRV. However, these differences in innate immune responses might not be critical for the regulation of viral growth in cells since three novirhabdoviruses showed similar proliferation in cells. It is well known that innate immune responses play a crucial part in the initiation and subsequent direction of adaptive immune response [100]. Thus, the differences in the expression of innate immune response genes by three novirhabdoviruses may affect the

efficiency of differentiation of appropriate adaptive immune responses required for protection against novirhabdoviruses and thus would have a connection to viral pathogenesis. Since both VHSV and HIRRV are pathogenic to flounder, the number of innate immune response genes altered by novirhabdovirus infection may not be critical to determine novirhabdovirus pathogenesis. Instead, specific genes/pathways affected by novirhabdovirus infection would determine viral pathogenesis. In this context, two groups of genes would play a critical role for connection to novirhabdovirus pathogenesis: 5 genes (2 genes, *TRIM46a* and *STAT1b*, up-regulated; 3 genes, *ADARb2*, *CCL19a.1*, and *TRIM2b*, down-regulated) which are commonly modulated more than two-fold by pathogenic both VHSV and HIRRV but not by IHNV; 6 genes (*IL10*, *IL12Rb2*, *STAT1b*, *CCR10*, *CXCL19*, and *CXCR5*) which are down-regulated more than two-fold only by IHNV. In addition, even though both VHSV and HIRRV are pathogenic to flounder, the overall innate immune response genes affected by these two viruses were distinct from each other, suggesting that their pathogenic mechanisms and disease symptom would differ between them. Further studies on the role of these genes in novirhabdovirus infection may reveal why IHNV is non-pathogenic in flounder while it proliferates well in host cells.

In summary, we show that IHNV is competent to replicate in flounder cells, similar to VHSV and HIRRV, even though it cannot cause disease in flounders. RNA-seq analysis identified 6 (*IRF8*, *CXCR1*, *TIRAP*, *CH25H*, *CXCL11.5*, and *TLR2*) and 3 (*CCR6a*, *IL-12a*, and *TLR1*) innate immune response genes commonly up-regulated and down-regulated in flounder cells, respectively, by all three novirhabdoviruses. Even though their antiviral activities against novirhabdovirus infection have not been determined, these genes, individually or collectively, may have important roles in the restriction of novirhabdovirus infection. However, our results suggest that the ability of novirhabdovirus to modulate the expression of these genes in virus-infected cells may not be the critical determinant for novirhabdovirus virulence in flounder. Consistently, it has been reported that IHNV virulence does not correlate with inhibition of IFN response in IHNV-infected fish [47].

Previously, many genome-wide analyses have been conducted to determine immune response to novirhabdoviruses in olive flounder [86, 99, 101–107], rainbow trout (*Oncorhynchus mykiss*) [108], sockeye salmon (*Oncorhynchus nerka*) [109], and turbot (*Scophthalmus maximus*) [86] and identified a large number of immune response genes modulated by novirhabdovirus infection. Of the 9 genes commonly modulated by three novirhabdoviruses, consistent with our data, genome-wide analyses identified TLR1 [93] and TLR2 [80] as being modulated by VHSV infection. However, we could not detect the change in the expression of the other 7 genes from the previous genome-wide analyses of novirhabdovirus-infected fish. While we used a flounder cell line HINAE, most other studies used tissue samples containing heterogeneous cells from virus-infected fish. Thus, it is possible that the previous genome-wide analyses might preferentially detect innate immune responses exerted by immune cells within the tissues rather than by virus-infected cells.

Beside the innate immune response, adaptive immune responses such as neutralizing antibody-mediated humoral immune response and cell-mediated immune response play important roles in protecting fish from novirhabdovirus infection [33]. Further studies for the comparison of transcriptome changes in flounder infected by IHNV, VHSV, and HIRRV are required to understand why–despite a robust innate immune response–flounder cell lines fail to shut down the replication of IHNV and why IHNV is avirulent for flounder.

## Supporting information

**S1 Table. List of differentially expressed genes (DEGs) in mock- and IHNV-infected HINAE cells (mock vs IHNV).**
(XLSX)

**S2 Table. List of DEGs in mock- and VHSV-infected HINAE cells (mock vs VHSV).**
(XLSX)

**S3 Table. List of DEGs in mock- and HIRRV-infected HINAE cells (mock vs HIRRV).**
(XLSX)

**S4 Table. Annotation of unigenes from IHNV-, VHSV-, HIRRV-infected HINAE cells to functional GO terms.**
(XLSX)

**S5 Table. List of 170 DEGs involved in innate immune response in IHNV-, VHSV-, or HIRRV-infected HINAE cells (mock vs virus).**
(XLSX)

**S6 Table. Cycle threshold (Ct) values used to generate graphs in Fig 4.**
(XLSX)

## Author Contributions

**Conceptualization:** Miyoung Cho, Sung-Hee Jung, Jeong Woo Park.

**Data curation:** Kwang Il Kim, Unn Hwa Lee, Eun Young Min, Jeong Woo Park.

**Formal analysis:** Kwang Il Kim, Unn Hwa Lee, Miyoung Cho, Eun Young Min.

**Funding acquisition:** Miyoung Cho, Jeong Woo Park.

**Investigation:** Unn Hwa Lee.

**Project administration:** Sung-Hee Jung.

**Resources:** Kwang Il Kim, Sung-Hee Jung.

**Supervision:** Sung-Hee Jung, Jeong Woo Park.

**Validation:** Unn Hwa Lee.

**Writing – review & editing:** Jeong Woo Park.

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
