## [Decision Letter · Decision Letter 0]

9 Jul 2020

PONE-D-20-17834

Transcriptome analysis based on RNA-seq of common innate immune responses of flounder cells to IHNV, VHSV, and HIRRV

PLOS ONE

Dear Dr. Park,

Thank you for submitting your manuscript to PLOS ONE. After careful consideration, we feel that it has merit but does not fully meet PLOS ONE’s publication criteria as it currently stands. Therefore, we invite you to submit a revised version of the manuscript that addresses the points raised during the review process.

We look forward to receiving your revised manuscript.

Kind regards,

Maria del Mar Ortega-Villaizan

Academic Editor

PLOS ONE

Additional Editor Comments:

The manuscript is of interest, however, deep corrections should be carried out before being accepted for publication. Please follow reviewers comments and corrections.

2. In your Methods section, please provide additional details regarding the cell lines used in your study and ensure you have described the source.

For more information regarding PLOS' policy on materials sharing and reporting, see

https://journals.plos.org/plosone/s/materials-and-software-sharing#loc-sharing-materials

and for more information on PLOS ONE's guidelines for research using cell lines, see

https://journals.plos.org/plosone/s/submission-guidelines#loc-cell-lines

Reviewers' comments:

Reviewer's Responses to Questions

**Comments to the Author**

1. Is the manuscript technically sound, and do the data support the conclusions?

Reviewer #1: Yes

Reviewer #2: Yes

2. Has the statistical analysis been performed appropriately and rigorously? 

Reviewer #1: Yes

Reviewer #2: I Don't Know

3. Have the authors made all data underlying the findings in their manuscript fully available?

Reviewer #1: Yes

Reviewer #2: No

4. Is the manuscript presented in an intelligible fashion and written in standard English?

Reviewer #1: Yes

Reviewer #2: No

5. Review Comments to the Author

Reviewer #1: General comments

This paper presents a comparative study on three fish rhabdoviruses (IHNV, VHSV and HIRRV) infection of a flounder cell line. It is noteworthy that while VHSV and HIRRV are pathogenic to flounder, IHNV is not. However, the three viruses replicate efficiently in flounder cells. Transcriptomic/RNAseq analysis revealed some pathways and immune-related genes modulated by the viral infections. The three viruses share a number of DEGs in common, while there are many pathways that only show up in one of the viral infections. Overall, this work contributes with novel information to the field of fish virology, but in the end the authors seemed unable to provide an explanation on why IHNV does not cause disease to flounder. It is somehow unfortunate that the key question regarding the difference between the two pathogenic viruses (VHSV and HIRRV) and the non-pathogenic IHNV remains unanswered. The suggestion that IHNV can not evade the cell antiviral response is not convincing, since there is no hard evidence supporting this hypothesis. In fact, this paper shows that IHNV replication on flounder cells does not differ from the other two rhabdoviruses. It is likely that some factors involved in this phenomenon are missed in the cell culture approach. A follow up work in vivo would surely shed more light on this interesting topic.

Specific issues

Materials and methods. All three viruses were grown at 20ºC. Although IHNV can grow at 20ºC, such temperature is higher than the reported optimal replication temperature for IHNV, which is around 14ºC. Nevertheless, here efficient replication of IHNV at 20ºC is shown, so I guess it is alright.

One striking piece of data here is that VHSV affects many more (more than twice) immune-related genes than IHNV or HIRRV. This is a remarkable finding, not discussed in the paper. I believe that author should comment on this subject.

P18. “there is no report on the role of CH25H in virus infected fish cells” I don´t think this is correct. The authors may want to take a look at a couple of papers (1) Pereiro, P., et al. (2017). "Interferon-independent antiviral activity of 25-hydroxycholesterol in a teleost fish." Antiviral Res 145: 146-159. (2) Zhang, Y., et al. (2019). "Fish cholesterol 25-hydroxylase inhibits virus replication via regulating interferon immune response or affecting virus entry." Front. Immunol 10(doi: 10.3389/fimmu.2019.00322).

Generally speaking, data from “omics” techniques are not easy to fit into a concrete mechanistic model. In this sense, I understand that the two pathogenic viruses (VHSV, HIRRV) should have some degree of similarity, while the transcriptome of IHNV-infected cells would differ in a number of genes/pathways that would have a connection to viral pathogenesis. So perhaps the answer is in those genes with a>2-fold decrease only in IHNV-infected cells. (which are hardly mentioned in the discussion).

Reviewer #2: The manuscript title “Transcriptome analysis based on RNA-seq of common innate immune responses of flounder cells to IHNV, VHSV, and HIRRV “by Kwang Il Kim and colleagues presents a work about the study of the transcriptomic profile of HINAE cells infected with IHNV, VHSV and VHSV. The results indicate that innate immune response genes were commonly modulated by all three novirhabdoviruses.

Despite the interesting study, it is necessary to do deep corrections in order to be published. In addition, the manuscript needs a profound English correction. I have divided the correction in sections:

In Material and methods

- Indicate in material and methods the specie (Paralichthys olivaceus) in HIRAME cells.

-The name of EPC cells has to be change to fathead minnow epithelioma papulosum cyprinid as indicate in the paper Winton J, Batts W, deKinkelin P, LeBerre M, Bremont M, Fijan N.Current lineages of the epithelioma papulosum cyprini (EPC) cell line are contaminated with fathead minnow, Pimephales promelas, cells. J Fish Dis.2010) 33:701–4. doi: 10.1111/j.1365-2761.2010.01165.x

-Authors should indicate in material and methods that “h “means hours at least the first time it appears.

- I found diverse style in the way to name the commercial references in material and methods, for example some with only the name of the company and others with the name and place please unify them. In addition, you should revise them because I found some reactives products without the commercial reference such as fetal calf serum. Please correct them.

-Which analysis method do you used to calculate the expression of genes in the real time PCR? Maybe 2^-DDCT?. You should indicate it in material and methods.

-Indicate the name and reference of statistical programme used to do the analysis and also the programme where graphs were performs.

In the results figures and tables

-In section 3.1 I found the next sentence difficult to understand “At day 1 (24 h) p.i., although the titer of IHNV was 6.3 and 6.9-fold lower for 1 and 0.1 MOI than that of VHSV, it was 1.6 and 2.1-fold higher for 1 and 0.1 MOI than that of HIRRV (Fig. 1)”. Please rewrite the sentence.

-In section 3.2 indicate the reference of GSEA programme.

-In the section 3.4 the genes,” IRF8, CXCR1, TIRAP, CH25H, CXCL11.5, and TLR2, were increased; 3 genes, CCR6a,IL12a, and TLR1” should include the full name of the genes in the text because it is the first time that these genes appear in the manuscript. Please check all the manuscript in case there were another mistake.

-In the figure 1, the legend indicates that “viral titers were determined in duplicate” the mean and standard deviation should be included in the graph results and indicate in the legend. In addition, the quality of the image is not good; the titles of the X and Y axes are a little bit blurry. Please, improve the image.

-In figure 2 the graph of HIRRV should have the same scale as VHSV and IHNV. Please correct it. In addition, it is really surprising that all the graphs are almost identical. I try to check the raw data in supplementary table S1,2 and3 however it is difficult to see the genes that belong to each pathway, can you do another table where only appear the raw data of the graphs ?.

- In figure 3 A, the graph represents a huge amount of genes so it is impossible to identify each gene because the title is so tiny and it is no possible to read. To solve the problem I do not know if increasing the size of the graph could improve the problem. Maybe the solution is only representing a few important data in the graph. Moreover, the reference of Venn diagrams should be included in the legend.

-In figure 4 is represented the expression of genes at 12 and 24 hours, but in the results there is not commented the 12 hours data. In my opinion these data should be commented or removed from the graphs. Moreover, I think it is important to have a supplementary table with the Ct data from real time PCR.

-In figure 5C and D, some circles cover the nodes of the image. Please correct them.

-In table 2: there is not reference or accession number of the primers, please add them.

- In table 4, I am not sure if the symbol “ # of genes” is a mistake. In the case that it was a mistake, please correct it.

- Table 5 should include in the legend the full name of every gene mentioned in the table.

In the discussion

The discussion needs a deep review. I miss a more argued discussion in general.

For example, I miss in the discussion a comparative between these transcriptomic results with other transcriptomic analysis with other fish infected with VHSV, IHNV and a microarray analysis with HIRRV.

Apart from that , I found the next sentences:

-This sentence “In addition, unigenes could be mapped to 55 KEGG pathways, indicating that novirhabdovirus infection modulates the expression of cellular genes involved in a broad variety of pathway”.it should be included in the results.

-“As in IRF8, the other genes in the group of six that were up-regulated, CXCR1, TIRAP,

CH25H, CXCL11.5, and TLR2, have been reported to be induced by virus infection and have

both antiviral and proviral functions”. This sentence should include a reference.

-“Binding of CXCL8/IL-8 to CXCR1 induces inflammation by recruiting immune cells to the site of damage and infection [60] and co-injection of IL-8 with VHSV G-encoding DNA vaccine in

rainbow trout enhanced the production of proinflammatory cytokine [61]. This suggests the

possibility that CXCL8/IL-8 induced by virus may enhance antiviral immunity. However, there

are several reports indicating that CXCL8/IL-8 inhibits the antiviral immunity of the host. IL-

8 induced by virus infection inhibits IFN-α activity in a dose-dependent manner [62] and IL-8

enhances the replication of several viruses including CMV [63], encephalomyocarditis virus

(EMCV), and poliovirus [62]”. This paragraph should be better explain, it is confusing.

6. PLOS authors have the option to publish the peer review history of their article (what does this mean?). If published, this will include your full peer review and any attached files.

Reviewer #1: No

Reviewer #2: No

---

## [Author Response · Author response to Decision Letter 0]

19 Aug 2020

I prepared responses to each point raised by editor and reviewers and enclose them with "Response to Reviewers" file.

---

## [Decision Letter · Decision Letter 1]

27 Aug 2020

PONE-D-20-17834R1

Transcriptome analysis based on RNA-seq of common innate immune responses of flounder cells to IHNV, VHSV, and HIRRV

PLOS ONE

Dear Dr. Park,

Thank you for submitting your manuscript to PLOS ONE. After careful consideration, we feel that it has merit but does not fully meet PLOS ONE’s publication criteria as it currently stands. Therefore, we invite you to submit a revised version of the manuscript that addresses the points raised during the review process.

We look forward to receiving your revised manuscript.

Kind regards,

Maria del Mar Ortega-Villaizan

Academic Editor

PLOS ONE

Additional Editor Comments (if provided):

The manuscript has significantly improved. However, some aspects should be corrected before being accepted. Please follow reviewer comments for it.

Reviewers' comments:

Reviewer's Responses to Questions

**Comments to the Author**

1. If the authors have adequately addressed your comments raised in a previous round of review and you feel that this manuscript is now acceptable for publication, you may indicate that here to bypass the “Comments to the Author” section, enter your conflict of interest statement in the “Confidential to Editor” section, and submit your "Accept" recommendation.

Reviewer #1: All comments have been addressed

Reviewer #2: All comments have been addressed

2. Is the manuscript technically sound, and do the data support the conclusions?

Reviewer #1: Yes

Reviewer #2: Yes

3. Has the statistical analysis been performed appropriately and rigorously? 

Reviewer #1: Yes

Reviewer #2: Yes

4. Have the authors made all data underlying the findings in their manuscript fully available?

Reviewer #1: Yes

Reviewer #2: Yes

5. Is the manuscript presented in an intelligible fashion and written in standard English?

Reviewer #1: Yes

Reviewer #2: Yes

6. Review Comments to the Author

Reviewer #1: (No Response)

Reviewer #2: In general authors have improved the manuscript substantially and they have corrected and answered my question properly. However, I still find mistakes that I consider that they should be corrected to be published.

- In general, the quality of the image of all the figures should be improved. They are blurred and this is the second time that I asked for an improvement in the figures.

- Legend figure1 should indicate that the mean and standard deviation was performed and also indicate the number of samples used in the experiment.

- In figure 1 the size of the titles of X and Y axes are not the same please correct them.

- In figure 2 the title of IHNV is in gray and VHSV and HIRRV are in black. Please unifies the title color.

- In the discussion authors have added a new sentence “In this context, two groups of genes would play a critical role for connection to novirhabdovirus pathogenesis: 5 genes (3 genes up-regulated;2 genes down-regulated) which are commonly modulated more than two-fold by pathogenic both VHSV and HIRRV but not by IHNV; 6 genes which are down-regulated more than twofold

only by IHNV”. It is difficult to know which genes are up or down regulated so I think you should identify these genes in order to clarify this sentence.

7. PLOS authors have the option to publish the peer review history of their article (what does this mean?). If published, this will include your full peer review and any attached files.

Reviewer #1: No

Reviewer #2: No

---

## [Author Response · Author response to Decision Letter 1]

3 Sep 2020

We responded to every comments raised by reviewer #2 in"Response to Reviewers" file.

---

## [Decision Letter · Decision Letter 2]

16 Sep 2020

Transcriptome analysis based on RNA-seq of common innate immune responses of flounder cells to IHNV, VHSV, and HIRRV

PONE-D-20-17834R2

Dear Dr. Park,

We’re pleased to inform you that your manuscript has been judged scientifically suitable for publication and will be formally accepted for publication once it meets all outstanding technical requirements.

Kind regards,

Maria del Mar Ortega-Villaizan

Academic Editor

PLOS ONE

Reviewers' comments:

Reviewer's Responses to Questions

**Comments to the Author**

1. If the authors have adequately addressed your comments raised in a previous round of review and you feel that this manuscript is now acceptable for publication, you may indicate that here to bypass the “Comments to the Author” section, enter your conflict of interest statement in the “Confidential to Editor” section, and submit your "Accept" recommendation.

Reviewer #2: All comments have been addressed

2. Is the manuscript technically sound, and do the data support the conclusions?

Reviewer #2: Yes

3. Has the statistical analysis been performed appropriately and rigorously? 

Reviewer #2: Yes

4. Have the authors made all data underlying the findings in their manuscript fully available?

Reviewer #2: Yes

5. Is the manuscript presented in an intelligible fashion and written in standard English?

Reviewer #2: Yes

6. Review Comments to the Author

Reviewer #2: Authors have addressed all my comments succesfully so I think that this paper is ready to be published

7. PLOS authors have the option to publish the peer review history of their article (what does this mean?). If published, this will include your full peer review and any attached files.

Reviewer #2: No

---

## [Editor Report · Acceptance letter]

18 Sep 2020

PONE-D-20-17834R2 

Transcriptome analysis based on RNA-seq of common innate immune responses of flounder cells to IHNV, VHSV, and HIRRV 

Dear Dr. Park:

I'm pleased to inform you that your manuscript has been deemed suitable for publication in PLOS ONE. Congratulations! Your manuscript is now with our production department. 

Kind regards, 

on behalf of

Dr. Maria del Mar Ortega-Villaizan 

Academic Editor

PLOS ONE